# Effects of Increasing Levels of Total Tannins on Intake, Digestibility, and Balance of Nitrogen, Water, and Energy in Hair Lambs

**DOI:** 10.3390/ani13152497

**Published:** 2023-08-02

**Authors:** Fabrício da Silva Aguiar, Leilson Rocha Bezerra, Maiza Araújo Cordão, Iara Tamires Rodrigues Cavalcante, Juliana Paula Felipe de Oliveira, Romilda Rodrigues do Nascimento, Bonifácio Benício de Souza, Ronaldo Lopes Oliveira, Elzania Sales Pereira, José Morais Pereira Filho

**Affiliations:** 1Graduate Program in Animal Science and Health, Federal University of Campina Grande, Patos 58708110, PB, Brazil; fabricio.aguiar18@hotmail.com (F.d.S.A.); romildarn01@ufpi.edu.br (R.R.d.N.); bonif@cstr.ufcg.edu.br (B.B.d.S.); 2Animal Science Departament, Faculdades Nova Esperança-FACENE, João Pessoa 58067698, PB, Brazil; maizacordao@hotmail.com; 3National Institute of Semi-Arid, Ministry of Science, Technology, and Innovations, Campina Grande 58434700, PB, Brazil; iara.cavalcante@insa.gov.br; 4Department of Animal Science, Federal University of Sergipe, Nossa Senhora da Glória 49680000, SE, Brazil; jupaula.oliv@yahoo.com.br; 5Department of Animal Science, Federal University of Bahia, Salvador 40170115, BA, Brazil; ronaldooliveira@ufba.br; 6Department of Animal Science, Federal University of Ceara, 2977, Mister Hull Avenue, Fortaleza 60356000, CE, Brazil; elzania@hotmail.com

**Keywords:** antioxidant substance, *Mimosa tenuiflora*, nitrogen, phenolic compounds, ruminant

## Abstract

**Simple Summary:**

Sheep farming plays a prominent role in the agriculture and economy of tropical regions, including the Brazilian semi-arid region. Rural producers have adopted *Mimosa tenuiflora* hay as a feed option for sheep and goats due to its prevalence in these areas. However, it is important to point out that *Mimosa tenuiflora* hay contains tannins, compounds with antinutritional and astringent properties. Therefore, it is crucial to understand the interaction of natural tannins derived from legumes with a ruminal metabolism in terms of energy and water balance. The inclusion of *Mimosa tenuiflora* hay as a source of tannins in the diet of hair lambs can potentially have positive effects on animal consumption.

**Abstract:**

This study aims to evaluate the effects of increasing tannin levels from *Mimosa tenuiflora* hay on the intake, digestibility, and balance of nitrogen (N), water, and energy in hair lambs. Thirty-two Santa Ines lambs, at an average age of 150 days and body weight of 26.75 ± 2.29 kg, were randomly assigned to four treatments in a completely randomized design. The treatments consisted of four diets: a control diet, tannin-free, and three diets with increasing levels of total tannin, 26.2, 52.4, and 78.6 g tannin/kg dry matter (DM). Including the total tannins in the lambs’ diet led to a quadratic increase in the intake of nutrients, N-retention (g/day), water intake, water absorption and retention, energy intake, and energy excretion in feces and gases. However, the digestibility of crude protein, neutral and acid detergent fibers, and total carbohydrates decreased. It was observed that there is a correlation between the variable nutrient digestibility and N-ingested and the N-absorbed, N-urinary, and N-retained. However, the N-excreted in feces did not correlate with any of the variables studied. It is recommended to include 33 g/kg DM of total natural tannins from *Mimosa tenuiflora* hay in the diet of hair lambs, as it improves intake, energy balance, dietary N, and body water composition while reducing the excretion of N-urinary and gas emissions to the environment.

## 1. Introduction

Sheep farming plays a significant role in the agriculture and economy of tropical regions. Sheep serve as an important source of animal protein for the local population, and their meat is valued for its taste and nutritional quality, making it a healthy food option. However, raising lambs, particularly in semi-arid areas, can be challenging due to the limited availability of natural or cultivated pastures. Thus, a major challenge in ruminant production is ensuring a consistent feed supply for animals that can be efficiently utilized. In this context, the production of hay during the rainy season for year-round use in total rations has become an increasingly necessary practice.

*Mimosa tenuiflora* is a deciduous shrub legume that is notable for its role in the diet of goats, sheep, and cattle. During the rainy season, these animals graze on its green leaves and branches, while during the dry season, they consume its pods, flowers, leaves, and dry branches. Due to these characteristics and its prevalence in the Brazilian semi-arid region, farmers have been using *Mimosa tenuiflora* hay as an alternative feed for sheep and goats. In a study conducted by Bandeira et al. [1], the average values of crude protein (CP), neutral detergent fibers (NDF), and acid detergent fibers (ADF) in *Mimosa tenuiflora* hay were found to be 14.4%, 44.5%, and 29.5%, respectively, whereas the digestibility of dry matter (DM) varies between 40% and 60% depending on the physiological phase and plant part, as observed by Pereira Filho et al. [2] and Cordão et al. [3].

In contrast, the *Mimosa tenuiflora* hay presents varying concentrations of tannins, ranging from 0.67 g/kg [4] to 1.8 g/kg DM [5]. These tannins can affect the intake and digestibility of dry matter [6], crude protein [7], and carbohydrates [8]. Due to their astringent properties, the presence of tannins can negatively affect water intake. They interact with salivary glycoproteins [9], causing them to precipitate, resulting in a loss of lubricating capacity [10].

Tannins are phenolic compounds that can be classified into hydrolyzable and condensed forms. They have a high molecular weight [11] and a strong capacity to form complexes with other substances [12], particularly proteins, through hydrogen bonding [13] and hydrophobic interactions. When tannins are added to the diet at levels exceeding 50 g condensed tannin/kg DM, they can reduce the efficiency of nitrogen utilization in the rumen [5,14,15], thereby reducing microbial protein production. However, when added in moderate levels, tannins can facilitate the passage of amino acids to the small intestine, thereby providing compensation [16]. The effects of tannins on sheep can vary depending on their concentration in the diet. Studies have reported positive, negative, or no significant changes in response [17]. However, at levels between 10 and 50 g tannin/kg DM, the literature highlights their beneficial role as antioxidants [18,19].

The intake of tannins by lambs at levels up to 50 g/kg DM has shown promising results. By protecting a portion of the protein from rumen bacteria, tannins can enhance intestinal digestibility [20]. They have an influence on the intake, digestion, absorption, and excretion of nitrogen [21], resulting in increased nitrogen retention in the ruminant’s body, [22,23] which subsequently contributes to improved meat quality [24]. However, when consumed in high amounts, tannins can be toxic [25] and can even lead to animal death [26,27]. The presence of condensed tannins >100 g/kg of DM improves the DM digestibility in goats and sheep, which might be associated with the adaptation of the animals to condensed tannins and because of the digestive physiology which appears to be associated with lower retention of ingested feeds [28]. The prolonged use of condensed tannins from *Mimosa tenuiflora* hay in sheep diets reduces body weight gain [15]. Therefore, it is crucial to understand how natural tannins derived from legume hay, with other factors such as the fiber fraction, may interact with the rumen metabolism of nitrogen, energy, and water balance. This is particularly relevant as animals in tropical regions often face limited availability of both forage and water under natural conditions.

Then, we hypothesized that the inclusion of tannins up to 7% DM in the total diet of lambs can promote protein protection and make it available in the small intestine, reducing mainly energy losses from the nitrogen metabolism and improving digestion and the nitrogen, energy, and water metabolism. In addition, it is crucial for enhancing the N-retention, water, and energy within the body, particularly in regions with limited water availability and vegetation abundant in tannin-rich plants. Therefore, the objective of this study was to assess the effects of increasing levels of tannins from *Mimosa tenuiflora* hay on the intake, digestibility, and the balance of nitrogen, water, and energy in hair lambs.

## 2. Materials and Methods

### 2.1. Tannins Source, Animals, and Experimental Diets

*Mimosa tenuiflora* hay was harvested from plants at the full vegetation stage in an area of the Caatinga biome, located in the northeast of the state of Paraiba, Brazil (latitude 07°04′49.68″ S, longitude 37°16′22.85″ W, and altitude of 264 m). The temperature ranged from 36.14 °C to 22.99 °C, with average air relative humidity of 56.92%.

To produce the hay, *Mimosa tenuiflora* branches of up to 7 mm in diameter were pruned, and then the material was shredded and hayed in the field. To achieve hay uniformity and minimize animal selection, the material was ground using a 2-cm sieve and subsequently bagged. Hay samples were collected for tannin analysis.

The phenolic compounds and tannin contents were determined by extracting phenolic compounds from the samples [27] using the 70% acetone method. The levels of total phenols, total tannins, and condensed tannins were measured using the Folin Ciocalteu method for total phenols, polyvinylpolypyrrolidone (PVPP) for total tannins, and Butanol-HCl for condensed tannins. After analyzing the tannin content of *Mimosa tenuiflora* hay, experimental treatments were established based on the following levels: 0 (control), 26.2, 52.4, and 78.6 g of total tannins per kilogram of dry matter. These levels corresponded to 0 (zero), 200, 400, and 600 g of hay per kilogram of dry matter in the total diet.

*Mimosa tenuiflora* and Buffel grass (*Cenchrus ciliaris*) hay were utilized as roughage (60%), while the concentrate (40%) was composed of ground corn and soybean meal (Table 1). The experimental diets were formulated to fulfill the necessary requirements for achieving a daily weight gain of 200 g, as recommended by the NRC [29].

Thirty-two crossbred Santa Inês male lambs, at an average age of 150 days and an initial body weight of 26.75 ± 2.29 kg, were used in the study. The lambs were initially weighed, identified, dewormed, and vaccinated against clostridium disease. Subsequently, the animals were placed in metabolic test cages, measuring 1.60 m in length, 0.80 m in width, and 1.50 m in height. Each cage was equipped with feeding and drinking trough. The diets were offered to the lambs at 0700 and 1600 h, with daily adjustments made to allow for 10% orts.

The animals underwent a 29-day adaptation period, followed by 7 days of sample and data collection. Individual samples of leftover feed, feces, and urine were collected and prepared for each animal daily during the experiment. The samples were then packed in containers and stored at −20 °C for subsequent chemical analysis.

The samples of feed, orts, and feces were pre-dried in forced ventilation ovens at 55 °C for 72 h. After pre-drying, the samples were ground and subjected to chemical analysis of dry matter content (DM—methods 967.03), mineral matter (MM—method 942.05), crude protein (CP—method 981.10), and ether extract (EE—method 920.29), according to AOAC [30]. For the determination of neutral detergent fibers (NDF) and acid detergent fibers (ADF), the recommended procedure by Van Soest [31] was followed, including the appropriate corrections for ash and protein content (_ap_NDF and _ap_ADF) and using thermostable α-amylase to remove starch.

The methodology of Sniffen et al. [32] was used to obtain the content of total carbohydrates (TCHO) and non-fiber carbohydrates (NFC), through the equations: TCHO = 100 − (CP + EE + CA) and NFC = 100 − (NDF + CP + EE + CA). Whereas for total digestible nutrients (TDN), the following equation was used: TDN = CP_DC_ + EE_DC_ × 2.25 + NDF_DC_ + NFC_DC_; and the crude energy was determined in calorimetric pump.

Digestibility coefficients (DC) of dry matter (DM), crude protein (CP), organic matter (OM), neutral detergent fibers (NDF), acid detergent fibers (ADF), neutral detergent fibers corrected for ash and protein (_ap_NDF), acid detergent fibers corrected for ash and protein (_ap_ADF), total carbohydrates (TCHO), and non-fiber carbohydrates (NFC) were calculated using the following equation: DC = (kg nutrient intake − kg nutrient excreted)/kg nutrient intake × 100.

Daily total urine collection was performed before feeding, using a plastic container with 10 mL of 1 N hydrochloric acid solution to prevent nitrogen loss. Aliquots of 10% of the daily volume were taken and stored in plastic bottles at −20 °C for subsequent chemical analysis in the Animal Nutrition Laboratory of UFCG.

### 2.2. Nitrogen (N) Balance

After quantifying nitrogen in urine using the Kjeldahl method, similar to the analysis performed on the diet and stool samples, the nitrogen balance was determined. The following equations were used to calculate the balance: N absorbed = N ingested − N fecal; N retained = N ingested − (N fecal + N urinary). These values were expressed in g/day and in function of metabolic weight.

### 2.3. Water Balance

The water balance was determined by evaluating the intake and excretion of water by the animals. To measure water intake, the weight of the water was recorded at the beginning and 24 h after it was offered. To account for evaporation, five buckets of water, similar to those used as drinking fountains, were placed randomly in the same area as the animal cages, and their weight was also recorded at zero and 24 h. The correction for evaporation was calculated by subtracting the weight of the evaporated water. To calculate metabolic water, the recommendations of Taylor et al. [33] and Church [34] were followed. The intake of digestible carbohydrates, protein, and ether extract were multiplied by factors of 0.60, 0.42, and 1.10, respectively. The following equations were used to calculate the water balance: Total water intake (TWI in g/day) = (water offered − water evaporated) + water contained in the diet + metabolic water; Total water excretion (TWE in g/day) = water excreted in urine + water excreted in feces; Absorbed water (AWA in g/day) = (water offered − water evaporated) + water contained in the diet − water excreted in feces; Retained water (RWA in g/day) = TWI − TWE; and Water balance (WBA in %) = (RWA/TWI) × 100.

### 2.4. Energy Balance

To determine the energy balance, samples of the diet, leftovers, feces, and urine were analyzed for their gross energy value using combustion in a calorimetric pump. Based on these results, the following calculations were performed: Digestible energy = Gross energy ingested − Gross energy of feces; and Metabolizable energy = digestible energy − gross energy lost in urine and gases. Meanwhile, gas energy was estimated as 8% of the total energy [35].

### 2.5. Statistical Analysis

A completely randomized design was used in this study with four treatments (tannin levels) and eight replications (lambs). The collected data were analyzed using the SAS statistical software (version 9.1) [36]. Prior to the analysis, the normality of the data was assessed using the Shapiro–Wilk test, and the homogeneity of variances was tested using the Bartlett test. Regression analysis was then conducted at a significance level of 5%. To compare the tannin-free diet with the diets containing tannins (0 × Tan), contrasts were performed, disregarding the specific tannin levels in the diet.

Pearson correlation analysis was conducted to examine the relationships between variables related to nutrient digestibility and nitrogen balance. After testing the relationships among the variables, the data went through principal component analysis (PCA) using the Past Program Software [37]. The PCA followed the methodology outlined by Ter Braak and Prentice [38].

## 3. Results

### 3.1. Intake and Nutrient Digestibility

The inclusion of tannins from *Mimosa tenuiflora* hay in the lambs’ diet resulted in a quadratic increase in the intake of all nutrients (*p* < 0.05), with coefficients of determination ranging from 0.49 for CP intake to 0.95 for NFC intake (Table 2). The inflection point, representing the maximum tannin level associated with the highest intake of each nutrient, varied across the nutrients. The highest estimated value was 45.1 g of tannins per kg of diet DM for NFC intake. On the other hand, the lowest observed value was 7.86 g of tannins in the diet for _ap_NDF intake. For DM and CP, the model estimated maximum values of 29.8 and 26.3 g tannins per kg of DM, respectively. When comparing the tannin-free diets with tannins (0 × Tan), it was observed that the tannin-free diet resulted in higher intakes (*p* < 0.05) of CP, _ap_NDF, _ap_ADF, NFC, and TDN. However, there were no significant differences (*p* > 0.05) in the intakes of DM, OM, and total carbohydrates.

The addition of tannins in the diet resulted in a linear decrease *(p* < 0.05) in the digestibility coefficients of OM, CP, _ap_NDF, _ap_ADF, and total carbohydrates, while the digestibility of NFC increased linearly (Table 3). The DM digestibility showed a quadratic increase (*p* < 0.05), reaching a maximum value of 63.6% with the addition of 17.4 g tannin/kg DM. When comparing the diets with and without tannin, it was observed that the tannin-free diet resulted in higher (*p* < 0.05) digestibility coefficients of OM, CP, _ap_NDF, _ap_ADF, and total carbohydrates, while a lower digestibility of NFC was observed. The DM digestibility was similar (*p* = 0.0569) between the two diets.

### 3.2. Nitrogen Balance

The N-ingested and N-excreted in the feces increased at initial levels of tannins inclusion, followed by a decrease to the level of 78.6 g/kg DM, characterizing a positive quadratic effect (*p* < 0.05) of tannins in the lambs’ diet, which allows one to estimate the maximum intake in g/day and in g/kg^0.75^ at the levels of 29.22 and 34.0 g of tannin/kg DM. Meanwhile, for that excreted in the feces, the model estimated 47.6 g of tannins per kg of diet DM (Table 4). The N-urinary excretion decreased linearly with the increasing tannin levels in the diet.

In terms of nitrogen balance, the absorption of nitrogen showed a negative linear decrease in g/day, while the retention of nitrogen in g/day showed a quadratic decrease. However, the tannin levels did not have a significant effect (*p* > 0.05) on the metabolic weight and the proportion of retained nitrogen in relation to the absorbed and ingested nitrogen. When comparing diets without and with tannins (0 × Tan), it was observed that animals on tannin-free diets presented a lower amount (*p* < 0.05) of N-excreted in feces, but a greater concentration of the N-urinary excretion and N-absorbed.

The principal components analysis applied to the dataset of digestibility and nitrogen balance revealed that the first two components (PC1 and PC2) account for 93% of the total variance. PC1 explains 81% of the variation and shows a strong correlation (R^2^ = 0.89 to 0.99) with the nutrient digestibility coefficients (Table 5). In PC2, which explains 12% of the total variation, there is a high correlation (R^2^ = 0.99) with the nitrogen excreted in the feces, as well as negative correlations (R^2^ = −0.42 and −0.35) with the urinary and absorbed nitrogen, respectively.

When examining the projection of variables on the principal components plane (Figure 1), it is evident that in PC1, all variables are projected in the positive quadrant, except for NFEC, which is closer to neutrality. In PC2, NINT and particularly NFEC stand out with a positive projection, contrasting with the negative projection of the NU and NAB variables.

### 3.3. Water Balance

The inclusion of the total tannins from *Mimosa tenuiflora* hay in the lambs’ diet resulted in a quadratic increase (*p* < 0.05) in all forms of water intake (from drinking and feed). However, the overall water intake in relation to DM intake was not affected by the tannin level (*p* > 0.05) in the diet (Table 6). There was no difference between the inclusion or not (0 × Tan) of tannins for the water intake source. The maximum estimated total water intake in kilograms and grams per kilogram of metabolic weight occurred when 26.54 and 32.80 g of tannins per kilogram of DM were reached, respectively. There was no effect of the tannin levels (*p* > 0.05) added to the lambs’ diet or difference between the inclusion or not of tannins (*p* > 0.05) on water excretion by the lambs.

Regarding water balance, there was a quadratic increase in water absorption and retention, both in quantity and in relation to the metabolic weight, with increasing tannin levels in the diet (*p* < 0.05). This suggests that higher levels of tannins (around 25.86 and 31.33 g per kilogram of DM) resulted in greater water retention in the lambs. However, when comparing the tannin-free diet with the diet containing tannins (0 × Tan), no significant difference (*p* > 0.05) was observed for any variable related to water balance.

### 3.4. Energy Balance

The inclusion of natural tannins from *Mimosa tenuiflora* hay in the lambs’ diet resulted in a quadratic increase (*p* < 0.05) in energy intake, energy excretion in feces, gas excretion, and the proportion of energy excreted in urine relative to the total energy intake. Additionally, there was a linear and increasing response in the percentage of energy excreted in feces and the energy intake (Table 7).

Regarding energy balance, the tannin levels had a quadratic effect on the digestible and metabolizable energy (Mcal/day and Mcal/kg^0.75^). However, the proportion of these values relative to energy intake decreased as the tannin levels increased in the diet. The highest energy intake, fecal excretion, and gas estimation values were observed when lambs received diets containing tannin levels ranging from 26.2 g to 52.4 g per kg of DM. When comparing the tannin-free diets to those with tannin, it was observed that the energy excretion in the feces (Mcal/kg^0.75^, % of gross energy intake—GEI) was higher (*p* < 0.05) in the diets containing tannin. However, the energy excretion in the urine (Mcal/day, Mcal/kg^0.75^, and % of GEI), as well as the digestible energy (Mcal/day, % of GEI) and metabolizable energy (% of intake) were lower when tannins were included in the lambs’ diets.

## 4. Discussion

The intake of all nutrients showed a positive quadratic response with the addition of tannins to the diet. The optimal inclusion levels of tannins were between 26.2 and 52.4 g tannin/kg DM, corresponding to an approximately 20 to 40% inclusion of *Mimosa tenuiflora* hay in the total diet of the lambs. This can be explained by the binding action of tannins with high-biological-value proteins, protecting them from proteolytic bacteria in the rumen and facilitating their release and absorption in the small intestine [39]. Additionally, tannins also play a role in controlling protozoa populations and contribute to the microbial balance in the rumen, leading to improved energy and protein utilization [20].

The maximum intake of DM, CP, and TDN was estimated when the tannin content in the diet reached 29.79, 26.25, and 33.72 g/kg DM, respectively. These findings align with the statements made by Valenti et al. [40], who suggested that tannin levels of up to 40 g/kg DM can enhance nutrient intake in lambs. Incorporating tannin levels of up to 15 g per kg DM in ruminant diets has been found to increase the flow of amino acids and non-microbial nitrogen into the duodenum, thereby reducing deamination processes that occur in the rumen and resulting in decreased nitrogen excretion in the urine [41,42].

It is worth highlighting that the quadratic effect of tannins on the nutrient intake in sheep was characterized by an increase in intake when the tannin level was below 40 g/kg DM. A better balance was observed in the intake of CP and NFC within the range of 26.2 to 52.4 g of tannins per kg DM, and a similar balance was observed for the fiber portion at around 20 to 30 g of tannins per kg DM. This balance ensured a favorable ratio between energy and protein, promoting a balanced interaction between the proteolytic bacteria and those involved in the breakdown of different carbohydrates in the diet. This is evident when considering that the highest estimate of TDN intake occurred when the tannin content of the diet reached 33.72 g/kg DM.

The similar intake can be attributed to a sharp reduction in the intake of all nutrients when the tannin level in the diet exceeded 52.4 g/kg of DM. This observation supports the notion that tannins, when present at appropriate levels, contribute to the microbial balance in the rumen by exerting selective antimicrobial activities that enhance the population of beneficial bacteria in the abomasum and intestine while inhibiting the growth of pathogenic bacteria. However, it is important to note that Serra et al. [18] have reported that tannin levels above 50 g/kg DM in ruminant diets can initially lead to weight loss and may even result in the intoxication and death of the animals. Ter Braak and Prentice [38] conducted a study with a maximum level of 80 g/kg DM of tannin extract from *Acacia mearnsii* in lambs’ diets and found that levels higher than 40 g of extract were harmful. Additionally, [43] suggested a limit of approximately 50 g tannins per kg DM for dairy goats, beyond which the efficiency of rumen microbiota in digesting ingested DM and the overall digestive efficiency in the rumen may be reduced.

The quadratic effect of tannins on nutrient intake is only evident in the digestibility of DM, where there is a positive linear response for NFC and a negative for the other nutrients, which explains the observed quadratic effect on DM digestibility. Furthermore, it is important to note that, contrary to the increased intake, the presence of tannins decreased the digestibility of nutrients, except for DM, which remained unchanged, and NFC, which was higher in the treatments with tannins.

The reduction in digestibility of NDF and ADF confirms that phenolic compounds also interfere with cell wall components. According to McSweeney et al. [44], as the tannin content increases in ruminant diets, the digestibility of the fiber components decreases. This can occur through various mechanisms, including the complexing of polymers such as cellulose and hemicellulose, the inhibition of microorganisms that directly act on NDF and ADF, the formation of chelates with mineral ions necessary for the metabolism of these bacteria, or a combination of these factors. Studies in sheep [45] and cattle [46] have also reported the effects of tannins on fatty acid biohydrogenation.

Based on the results obtained for intake and nutrient digestibility, it is suggested that the impact of tannins in the rumen environment is primarily due to their interaction with enzymes, proteins, and carbohydrates, rather than their astringency [47]. This interaction may lead to a longer retention time of the fiber components (NDF and ADF) in the diet, resulting in delayed rumen emptying and potentially influencing feed intake control, which can go from a chemical regulation to a more physical regulation [48].

The quadratic effect observed of the tannin levels and nitrogen retention (g/day) indicates an increase in N-retained up to a tannin level of 26.2 g per kg DM. This can be explained by the linear reduction in N-urinary excreted, as well as the increased amounts of nitrogen ingested and excreted in the feces. However, at higher tannin concentrations, N-excretion in urine and feces decreased. These findings confirm that tannins, at appropriate levels, promote a favorable nitrogen balance by enhancing its retention in the lambs’ bodies. These findings are confirmed by Mueller-Harvey [49], who demonstrated that tannins, through the formation of stable complexes with proteins, decrease rumen degradation and significantly influence N-balance. N-urinary, in the form of uric acid, contributes to ammonia pollution in the environment [50], whereas fecal nitrogen can be converted into organic matter for the soil [51]. This behavior was observed in our study when comparing the treatments without tannins to those with tannins, wherein N-ingested was similar, but lambs fed tannin-containing diets showed lower N-urinary excretion and higher N-fecal excretion.

The utilization of tannins in ruminant diets is only justified when it leads to an improved production performance and the production outcomes retained in the animal’s body result in high-quality meat and/or milk for human consumption. In this context, the impact of tannins on nitrogen metabolism is noteworthy [20]. Tannins have been found to reduce proteolytic bacteria in the rumen and inhibit rumen biohydrogenation of fatty acids. As a result, there is an increase in the concentration of unsaturated fatty acids in the abomasum and intestine, leading to an improved muscle nitrogen balance, and contributing to the production of meat [52] with a higher proportion of unsaturated fats, which reduces human health risks.

PC1 explained 81% of the variation and showed a strong correlation between digestibility variables and NINT, NU, NAB, and NR, which can be attributed to the effect of tannins on NU (Figure 1). PC2, which explained 12% of the variation, was characterized by the lack of correlation between nutrient digestibility and NFEC, as well as the quadratic effect of tannin on N-fecal. The projections in the principal component analysis showed the distinct patterns between tannin-free treatments and the treatment with the highest tannin level (0 and 78.6), as well as the opposing projections observed for NFEC and NU/NAB in different quadrants.

The quadratic effect of tannins on the water intake, without affecting the amount of water excreted in urine and feces, suggests that lambs can maintain adequate levels of water in their bodies by reducing water excretion without compromising body homeostasis. Although the tannin levels influenced the nutrient intake and digestibility, the total water intake expressed as %DM ingested remained unchanged. This finding highlight that those tannins primarily influenced the intake rather than directly impacted the digestibility of the nutrients. This contradicts the assumption that changes in the dry matter intake necessarily led to changes in the water intake. Ruminants have mechanisms to reduce water loss in urine [53] and feces [54], as well as the increased production of metabolic water [55], which collectively contribute to the animal’s water balance [56].

The contrasting effect of tannins on the total water intake, and the similarity in the total water excretion, can be attributed to the higher intake of free water and dietary water, as well as increased metabolic water production in the diet with a low tannin level. However, at higher tannin levels, these values decrease, which suggests that lambs can regulate water retention by controlling its excretion [57]. This regulatory mechanism is important for maintaining homeostasis, as the osmolarity of body fluids plays a critical role. The renin–angiotensin–aldosterone system is involved in the control of water intake and excretion [58].

The quadratic effect of tannin on water retention follows a similar pattern as observed for nutrient intake, DM digestibility, and N retention in grams, always increasing when tannin levels are below 40 g/kg DM and decreasing at higher tannin levels. The decrease in the digestibility of CP, NDF, and NFC with increasing tannin levels may have contributed to the maximum water retention that was estimated at a tannin level of 31.33 g/kg DM, confirming the impact of protein and carbohydrate digestibility on the lambs’ water balance. This is evident when comparing the metabolic water production between the free-tannin diet (402.06 g) and the diet with the highest tannin level (78.6 g of tannin per kg DM), which decreased to 200.04 g, resulting in a 49.75% reduction in metabolic water production.

In ruminants, the variation in the nutrient intake in diets with different tannin concentrations is generally associated with astringency, which also affects water retention in the body. In this study, it was found that water balance was primarily influenced by nutrient intake rather than nutrient digestibility. It is worth noting that the effect of tannins on water intake in mass unit (g and/or kg), regardless of its source (free water or from the diet), did not have a similar effect on water excretion in feces, urine, or total, which suggests that lambs in semi-arid regions employ defense mechanisms to conserve water, as observed in other studies [59,60], where lambs retained or eliminated water to maintain their body fluids and ensure proper physiological functions. Similar observations were made by Magalhães et al. [61] in their evaluation of diets with different proportions of bean residues combined with cactus pear in hair lambs.

According to Karimizadeh et al. [62], a significant portion (60 to 90%) of the variations in nutrient retention in the animal’s body is attributed to nutrient ingestion, while nutrient digestibility accounts for a smaller percentage (10 to 40%), even when high-moisture diets are used [53]. However, regardless of astringency, the crucial factor is the amount of tannins ingested. At low levels, tannin can have beneficial effects, but at high levels, it has the potential to influence the water balance of ruminants, affecting both water intake and urine excretion.

It is crucial to determine the appropriate level of tannin for each specific ruminant species and category to ensure optimal animal performance without compromising organic functions. Studies conducted by Cordão et al. [3] and Bandeira et al. [1] with hair lambs recommend tannin levels of up to 40 g per kg of dry matter (DM) in the diet. Similarly, [42] observed that hydration and milk production in dairy cows can be maintained by incorporating up to 4.3 g of tannins per kg of DM in the diet.

The positive quadratic response of energy intake as a function of tannin content in the diet is consistent with the findings for intake and digestibility. This indicates that the inclusion of appropriate levels of tannins from *Mimosa tenuiflora* can influence intake, digestibility, and nutrient balance, potentially leading to improved animal performance and enhanced meat quality in lambs. According to Vasta et al. [63] tannins can affect the energy balance of ruminants, including their impact on rumen microbiology, such as reducing the biohydrogenation of unsaturated fatty acids. This effect has been observed by Gama et al. [64] in sheep and goats that were finished on native pastures enriched with buffel grass, where the presence of tannin-rich plants such as *Mimosa tenuiflora* in the grazing area was prominent.

When comparing the balance, intake, and excretion of energy between tannin-free diets and those with tannins (0 × Tan), the results further highlight the potential of *Mimosa tenuiflora* tannins to enhance or inhibit functions in a sheep’s body. The presence of tannins in the diet leads to a reduction in the digestible energy (Mcal/day and % of GEI) in lambs, likely due to an increased energy excretion in the feces, especially in animals receiving higher levels of tannins (52.4 and 78.6 g per kg of DM). Additionally, there is a linear decrease in the digestibility of _ap_NDF and a linear increase in the digestibility of NFC, indicating that the effects of tannins on intake and digestibility may differ depending on the tannin level in the diet. It is worth noting that the astringency and bitter taste caused by tannins have a primary effect on intake, while the impact on digestibility is secondary.

The selectivity behavior of lambs compensates for the negative effects of tannins on intake and digestibility. Lambs tend to prefer more palatable components that have a lower tannin content, and when that occurs, the effects on both intake and digestibility become similar, thereby promoting a better nutritional balance. This is beneficial because tannins reduce hydrogen (H_2_) production in the rumen, which inhibits the growth of methanogenic bacteria [9] and fermenters of cellulose and hemicellulose, decreasing the digestibility of fiber carbohydrates [63], leading to reduced methane production [65] and the release of energy in the form of pollutant gases into the environment [66].

However, at high levels of tannins, surpassing the natural selection capacity of sheep, they can disrupt the population balance of bacteria [67] and protozoa [68], and their prolonged use even leads to toxicity [20] and, potentially, cause the death of animals. In this study, the results of the energy balance analysis indicated that the optimal levels of tannins for maximizing digestible energy and metabolized energy were estimated to be 16.1 and 16.3 g per kg of dry matter, respectively. These values represent an average reduction of approximately 40% and 58% compared to the estimated levels of 27 and 39 g tannins, respectively, required to maximize energy intake and minimize energy excretion in the feces of lambs.

## 5. Conclusions

Using *Mimosa tenuiflora* hay as a tannin source in the diet of hair lambs up to a level of 33 g tannins per kg of DM, equivalent to 30% of the total diet, enhances the intake, N-retention, and body water in the lambs while reducing the excretion of urinary nitrogen and gases into the environment.

## Figures and Tables

**Figure 1 animals-13-02497-f001:**
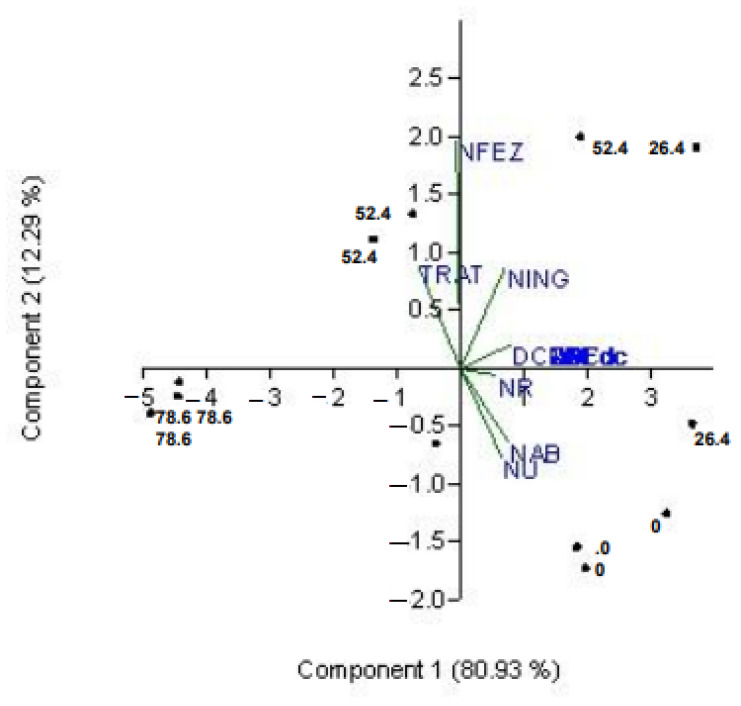
Projection of the variables on the principal components plane for, Coefficient of Digestibly (CD), N-intake (NINT), N-fecal (NFEC), N-urinary (NU), and N-absorbed (NAB).

**Table 1 animals-13-02497-t001:** Participation of ingredients in the ration and chemical composition of the experimental diets.

Ingredients (%)	Tannin Levels (g/kg DM)			
	0	26.2	52.4	78.6
*Cenchrus ciliaris* hay	60.00	40.20	19.80	0.00
*Mimosa tenuiflora* hay	0.00	19.80	40.20	60.00
Ground corn	24.17	25.08	24.63	26.83
Soybean meal	13.74	12.81	13.82	11.09
Urea	0.50	0.50	0.17	0.46
Soybean oil	0.23	0.24	0.00	0.22
Calcite Limestone	0.36	0.37	0.38	0.40
Mineral Miture ^1^	1.00	1.00	1.00	1.00
Chemical composition of the diet (g/kg DM)				
Dry matter (g/kg as fed)	916.60	914.50	911.60	909.80
Organic matter	912.97	925.74	943.76	952.31
Mineral matter	74.76	61.98	46.28	35.62
Crude protein	128.35	128.66	128.30	128.92
_ap_Neutral detergent fibers ^2^	517.53	495.89	477.63	452.33
_ap_Acid detergent fibers ^2^	330.13	329.15	330.29	327.36
Ether extract	28.69	30.96	32.31	35.30
Total carbohydrates	786.51	794.05	796.98	809.13
Non-fiber carbohydrates	268.97	298.15	319.33	356.79
Total digestible nutrients	613.22	613.80	617.89	615.46
Phenolic compounds	000	30.11	60.25	90.34
Total tannin	0.00	26.20	52.42	78.60
Condensed tannin	0.00	11.88	23.78	35.65
Water-soluble tannin	0.00	14.32	28.64	42.95

^1^ Mineral mix (nutrient/100 g of element): calcium 15.30 g; phosphorus 7.0 g; sodium 14.80 g; magnesium 0.13 g; sulfur 1.20 g; cobalt 14.0 mg; iodine 6.10 mg; manganese 396.0 mg; selenium 0.50 mg; zinc 470.0 mg; iron 220.0 mg. ^2^ Corrected for ash and protein.

**Table 2 animals-13-02497-t002:** Nutrient intake (g) of lambs fed different levels of tannins in the diet.

Intake (g)	Tannins (g/kg DM)				SEM	R²	*p*-Value		
	0	26.2	52.4	78.6			L	Q	0 × Tan.
Dry matter	1076.35	1396.32	1117.84	758.85	24.66	0.90	0.0244	0.0001	0.6163
Organic matter	995.87	1301.07	1043.73	721.25	22.92	0.89	0.0334	0.0001	0.3425
Crude protein	117.22	114.39	127.40	82.28	2.61	0.49	0.1581	0.0409	0.0014
_ap_Neutral detergent fibers	651.08	736.60	440.20	301.43	11.43	0.86	0.5518	0.0072	0.0001
_ap_Acid detergent fibers	463.94	538.32	323.03	270.52	8.22	0.68	0.0001	0.0486	0.0001
Total carbohydrates	857.77	1091.47	844.80	588.74	18.77	0.88	0.0070	0.0001	0.4718
Non-fiber carbohydrates	206.69	354.86	404.61	287.40	8.38	0.95	0.1092	0.0001	0.0001
Total digestible nutrients	581.74	799.14	708.42	439.73	15.28	0.95	0.1063	0.0001	0.0025

SEM = Standard Error of the Mean; R² = Coefficient of determination; L = Linear; Q = Quadratic; _ap_NDF and _ap_ADF = corrected for ash and protein.

**Table 3 animals-13-02497-t003:** Nutrient digestibility (g/kg DM) in lambs fed different levels of tannins in the diet.

Nutrient (g/100 g Ingested)	Tannins (g/kg DM)				SEM	R²	*p*-Value		
	0	26.2	52.4	78.6			L	Q	0 × Tan.
Dry matter	62.5	63.5	59.2	50.5	2.04	0.76	0.0406	0.0336	0.0569
Organic matter	65.1	65.1	59.1	50.6	1.99	0.71	0.0006	0.0513	0.0174
Crude protein	71.6	53.4	49.1	39.5	1.30	0.93	0.0001	0.0501	0.0001
_ap_Neutral detergent fibers	62.5	58.1	39.4	23.2	3.32	0.94	0.0001	0.0526	0.0001
_ap_Acid detergent fibers	63.5	57.2	36.0	27.8	1.57	0.93	0.0001	0.7185	0.0001
Total carbohydrates	65.2	64.7	59.6	51.6	1.96	0.70	0.0007	0.0746	0.0199
Non-fiber carbohydrates	73.7	78.6	82.1	81.5	1.53	0.57	0.0043	0.0900	0.0040

SEM = Standard Error of the Mean; R² = Coefficient of determination; L = Linear; Q = Quadratic; DM = Dry matter; OM =; CP =; apNDF and apADF = corrected for ash and protein.

**Table 4 animals-13-02497-t004:** Nitrogen balance in lambs fed different levels of tannins in the diet.

Variables	Tannins (g/kg DM)				SEM	R²	*p*-Value		0 × Tan.
	0	26.2	52.4	78.6			L	Q	
	Nitrogen Intake								
g/day	18.87	20.06	20.53	12.83	1.02	0.76	0.050	0.003	0.394
g/kg^0.75^	1.71	1.74	1.89	1.31	0.09	0.58	0.125	0.020	0.573
	Nitrogen Excretion								
Urine (g/day)	9.96	6.48	6.15	3.19	0.78	0.76	0.001	0.769	0.001
Urine (g/kg^0.75^)	0.90	0.56	0.57	0.32	0.07	0.73	0.001	0.550	0.001
Fecal (g/day)	5.64	8.52	12.05	8.08	0.87	0.65	0.126	0.008	0.005
Fecal (g/kg^0.75^)	0.51	0.74	1.11	0.82	0.07	0.68	0.032	0.018	0.002
	Nitrogen Balance								
Absorbed (g/day)	13.23	11.54	8.48	4.75	0.69	0.89	0.001	0.155	0.001
Retained (g/day)	3.26	5.06	2.33	1.57	0.41	0.58	0.054	0.021	0.569
Retained (g/kg^0.75^)	0.30	0.44	0.21	0.16	0.05	0.27	0.395	0.131	0.619
Retained (%absorbed)	24.51	44.85	27.59	33.25	6.25	0.01	0.332	0.348	0.225
Retained (%ingested)	17.26	25.52	11.07	12.03	2.70	0.16	0.690	0.364	0.730

SEM = Standard error of the mean; R^2^ = Coefficient of determination; L = linear; Q = quadratic.

**Table 5 animals-13-02497-t005:** Eigenvalues, variance of principal components, and correlations of the variables in each component.

Item	Component 1 (PC1)	Component 2 (PC2)
Eigenvalues	9.71	1.47
% of Variance	0.81	0.12
Accumulated Variance (%)	0.81	0.93
Variables		
Dry matter ^DC^	0.99	0.05
Organic matter ^DC^	0.99	0.05
Crude protein ^DC^	0.96	0.09
Neutral detergent fibers ^DC^	0.93	0.08
Acid detergent fibers ^DC^	0.91	0.16
Total carbohydrates ^DC^	0.96	0.14
Non-fibrous carbohydrates ^DC^	0.95	0.03
N-intake	0.89	0.41
N-fecal	−0.04	0.99
N-urinary	0.80	−0.42
N-absorbed	0.92	−0.35
N-retained	0.69	−0.05

^DC^ is digestibility coefficient.

**Table 6 animals-13-02497-t006:** Water balance in lambs fed different levels of tannins in the diet.

Variables	Tannins (g/kg DM)				SEM	R²	*p*-Value		0 × Tan
	0.0	26.2	52.4	78.6			L	Q	
	Water Intake Source								
Drinking (kg)	2.969	3.235	3.046	2.268	0.199	0.63	0.093	0.022	0.619
Feed (g)	93.56	103.17	100.27	70.18	5.51	0.72	0.069	0.005	0.721
Metabolic (g)	402.06	410.89	351.17	200.04	32.15	0.77	0.290	0.027	0.060
Total (kg)	3.464	3.749	3.497	2.538	0.229	0.67	0.096	0.018	0.465
Total (g/kg^0.75^)	296.23	325.16	322.58	259.09	16.53	0.56	0.250	0.017	0.759
Total (g/g DM)	3.09	3.09	3.10	3.37	0.13	0.28	0.160	0.305	0.544
	Water excretion								
Feces (g)	353.71	376.16	328.25	249.29	54.60	0.28	0.147	0.351	0.586
Feces (g/kg^0.75^)	30.42	32.55	30.32	25.41	4.71	0.12	0.622	0.449	0.861
Urine (g)	624.56	720.83	650.78	664.56	86.84	0.08	0.892	0.635	0.604
Urine (g/kg^0.75^)	53.34	62.61	60.04	67.89	7.81	0.14	0.672	0.927	0.292
Total (kg)	0.978	1.097	0.979	0.914	0.103	0.13	0.500	0.383	0.882
Total (g/kg^0.75^)	83.75	95.16	90.36	93.30	9.01	0.06	0.536	0.637	0.403
	Water balance								
Absorbed (kg)	2.709	2.962	2.818	2.088	0.19	0.60	0.091	0.024	0.707
Absorbed (g/kg^0.75^)	231.5	257.0	259.9	213.3	13.9	0.48	0.510	0.023	0.479
Retained (kg)	2.486	2.652	2.518	1.624	0.21	0.63	0.137	0.029	0.397
Retained (g/kg^0.75^)	212.5	230.0	232.2	165.8	16.7	0.53	0.161	0.029	0.397
Retained (%)	71.78	70.52	71.89	63.99	2.75	0.16	0.532	0.256	0.214

SEM = Standard error of the mean; R^2^ = Coefficient of determination; L = linear; Q = quadratic.

**Table 7 animals-13-02497-t007:** Energy balance in lambs fed different levels of tannins in the diet.

Variable		Tannins (g/kg DM)				SEM	R^2^	*p*-Value		0 × Tan
		0.00	26.2	52.4	78.6			L	Q	
		Gross energy intake (GEI)								
GEI (Mcal/day)		4.963	5.367	5.080	3.401	0.289	0.77	0.040	0.004	0.328
GEI (Mcal/kg^0.75^)		0.424	0.466	0.468	0.347	0.021	0.71	0.146	0.003	0.889
		Gross energy excretion (GEE)								
Feces (Mcal/day)		1.782	2.193	2.349	1.725	0.132	0.64	0.985	0.003	0.081
Feces (Mcal/kg^0.75^)		0.152	0.191	0.216	0.176	0.009	0.70	0.190	0.003	0.005
Feces (% of GEI)		35.96	40.76	46.57	50.78	2.03	0.79	0.001	0.881	0.003
Urine (Mcal/day)		0.154	0.096	0.099	0.079	0.017	0.32	0.070	0.246	0.008
Urine (Mcal/kg^0.75^)		0.014	0.008	0.009	0.008	0.001	0.40	0.099	0.212	0.021
Urine (% of GEI)		3.20	1.79	1.98	2.37	0.38	0.47	0.284	0.038	0.029
Gases (Mcal/day)		0.397	0.429	0.406	0.272	0.033	0.77	0.040	0.004	0.328
		Energy balance								
Digestible (Mcal/day)		3.181	3.175	2.731	1.676	0.219	0.80	0.393	0.032	0.032
Digestible (Mcal/kg^0.75^)		0.271	0.276	0.252	0.171	0.017	0.74	0.261	0.031	0.094
Digestible (% of GEI)		64.04	59.24	53.43	49.22	2.03	0.79	0.001	0.881	0.002
Metabolizable (Mcal/day)		2.625	2.649	2.225	1.324	0.195	0.79	0.388	0.033	0.038
Metabolizable (Mcal/kg^0.75^)		0.224	0.230	0.205	0.135	0.016	0.74	0.266	0.033	0.103
Metabolizable (%absorbed)		82.51	83.40	81.21	78.90	1.02	0.43	0.493	0.137	0.287
Metabolizable (%ingested)		52.84	49.45	43.45	38.85	2.09	0.76	0.002	0.768	0.006

SEM = Standard error of the mean; R^2^ = Coefficient of determination; L = linear; Q = quadratic.

## Data Availability

Data are not publicly available due to restrictions on the research group, but can be requested from the corresponding author.

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
