# Peer review of "Effects of Increasing Levels of Total Tannins on Intake, Digestibility, and Balance of Nitrogen, Water, and Energy in Hair Lambs"

_animals, 2023, doi:10.3390/ani13152497_

Round 1
Reviewer 1 Report
The article examines effects of increasing levels of tannins from Mimosa tenuiflora hay on intake, digestibility, and the balance of nitrogen, water, and energy in hair lambs, The trial has yielded significant results, serving as a valuable guide for production.
Some minor comments:
1. On what grounds were the three ratios of tannin employed in the trial chosen? The basis for the data selection remains unclear.
2. When applying linear analysis to the four treatment groups, how is the accuracy of the data ensured?
3. Lines 142/145: There is a need to confirm and potentially update the methodological basis of determination (AOAC).
4. Some of the formatting in the text needs to be checked again, e.g. line 248, g/kg0.75 must be superscripted.
5. Lines 330-332: It is recommended to provide specific grounds for the assertions made.
6. Line 360: Please confirm if reference 38, as cited here, is the correct one.
7. Line 514: Could you offer a detailed explanation concerning the optimal additions mentioned in the conclusion?
8. Some references appear to be incorrectly formatted and require correction, for instance, references 11 and 20.
9. The references require updating. It is recommended to cite more high-impact papers published within the last five years.
Author Response
Dear Animals-MDPI Editorial Office
We have thanked the adjustments suggested, and we appreciate the attention of the reviewer's contribution to the analysis and correction of this manuscript. We have modified the manuscript according to the review points and are forwarding our paper's updated version. All corrections were addressed, as shown below and in the attached file. Answers to the questions are provided below. All the manuscript changes have been highlighted in yellow for Reviewer #1.
Sincerely yours,
Dr. Leilson Rocha Bezerra

Reviewer 2 Report
Introduction
general suggestion
I recommend reassessing the introduction and making it clearer and more connected with the purpose of the study.
Please present the study hypothesis clearly
M&M's
It is not appropriate to use Pearson's correlation analysis to test variables for principal component analysis (PCA). What should be done are the sphericity tests and KMO = kaiser-meyer-okin.
Perform the PCA analysis with all the study response variables and use it to evaluate the variation of the variables so that you can present possible mechanisms to describe your results
Results
Table 5 = Review these values, correlation coefficients are all the same.
I recommend removing the correlation analysis, in addition to not adding importance to the discussion and much more interesting to use the PCA, it will bring greater gains to the work
Discussion
Overall = The discussion should be rewritten in order to make it more direct and focused on discussing the results of the study, focusing on the mechanisms. The discussion still presents a lot of presentation of results.
Within the discussion there should be several conclusions, such as what information your work brings to the scientific community.
For the variables that presented answers that quadratic effect it would be interesting to perform the integral function so you could find the maximum and minimum value
Line 330-332 = This hypothesis needs to be referenced
Line 350-357 = Confusing paragraph and still looking like results show
Author Response
Dear Animals-MDPI Editorial Office
We have thanked the adjustments suggested, and we appreciate the attention of the reviewer's contribution to the analysis and correction of this manuscript. We have modified the manuscript according to the review points and are forwarding our paper's updated version. We are, of course, always available, and we are pleased to provide any clarification that may be required. All corrections were addressed, as shown below and in the attached file. Answers to the questions are provided below. All the manuscript changes have been highlighted in blue for Reviewer #2 comments.
Sincerely yours,
Dr. Leilson Rocha Bezerra

Reviewer 3 Report
This paper discusses an important topic within tropical ruminant animal production, which can also have implications in other parts of the world. Tannins can be found in a number of feedstuffs and having more understanding of the threshold of their inclusion in the diet of ruminants, large or small, is important for ruminant nutrition. The major aim of this paper is identifying how the level of tannin inclusion impacts nutrient digestion and balance within small ruminants.
With regards to the literature review, I believe it is done very well and the authors did a good job of covering the relevant research on this topic. I did find 1 paper that might be worth considering for inclusion which is Naumann et al, 2017 (in R. Bras. Zootec.). There are a few questions I have regarding the paper as well. First, there are a couple of places, including the implications / conclusion, where "recommended" feeding values are presented, but there does not seem to be any supporting evidence in the paper as those values were not 1 of the 3 treatment levels tested; so how are those recommendations relevant given the presented data?? With regard to the Pearson correlation coefficients and subsequently variance of principal components (Tables 5 & 6), are all of those digestion coefficients correct, given that they are all the same?? Please double check your spacing on the "p-values" in the results sections for consistency, as some have a space between the "p" and the number while others do not.
Specific edits:
- Line 63, italicize "Mimosa tenuiflora"
- In the materials and methods, suggest changing "leftovers" to "orts"
- Line 134, change "7 am and 4 pm" to "0700 & 1600 h"
- Line 248 & 249, this is an example of numbers which are discussed, but are not shown as to where they came from...
- In Table 4, for the last row, would it be better defined by "Retained (%ing)" as opposed to "Retained (%reta)"
- Lines 261 & 262, the "0.89" and "0.77" following the p-values are undefined; are they r-values or what specifically?
- In Table 8, would it be a benefit to add units (% abs and % __) to the last two rows?
- Lines 358, 360, 362, & 473, add the Author's names before the reference number.
- Line 471, the "Bandeira" reference is listed as both # 1 & # 15; also be sure to address this in the References section
-
English use is very appropriate and generally well done.
Author Response
Dear Animals-MDPI Editorial Office
We have thanked the adjustments suggested, and we appreciate the attention of the reviewer's contribution to the analysis and correction of this manuscript. We have modified the manuscript according to the review points and are forwarding our paper's updated version. We are, of course, always available, and we are pleased to provide any clarification that may be required. All corrections were addressed, as shown below and in the attached file. Answers to the questions are provided below. All the manuscript changes have been highlighted in green for Reviewer #3 comments.
Sincerely yours,
Dr. Leilson Rocha Bezerra

Reviewer 4 Report
The manuscript shows relevant results that could be improved with some adjustments in the interpretation of some description and writing errors.

Author Response
Dear Animals-MDPI Editorial Office
We have thanked the adjustments suggested, and we appreciate the attention of the reviewer's contribution to the analysis and correction of this manuscript. We have modified the manuscript according to the review points and are forwarding our paper's updated version. We are, of course, always available, and we are pleased to provide any clarification that may be required. Answers to the questions are provided below. All the manuscript changes have been highlighted in purple for Reviewer #4 comments.
Sincerely yours,
Dr. Leilson Rocha Bezerra

Round 2
Reviewer 2 Report
no comments
Reviewer 3 Report
Thank you for your revisions and I believe this paper looks very good.